# Relationship between Mechanical Ventilation and Histological Fibrosis in Patients with Acute Respiratory Distress Syndrome Undergoing Open Lung Biopsy

**DOI:** 10.3390/jpm12030474

**Published:** 2022-03-16

**Authors:** Hsin-Hsien Li, Chih-Wei Wang, Chih-Hao Chang, Chung-Chi Huang, Han-Shui Hsu, Li-Chung Chiu

**Affiliations:** 1Institute of Emergency and Critical Care Medicine, School of Medicine, National Yang Ming Chiao Tung University, Taipei 11221, Taiwan; hsinhsien@mail.cgu.edu.tw (H.-H.L.); hsuhs@vghtpe.gov.tw (H.-S.H.); 2Department of Respiratory Therapy, Chang Gung University College of Medicine, Taoyuan 33302, Taiwan; cch4848@cgmh.org.tw; 3Department of Pathology, Chang Gung Memorial Hospital, Chang Gung University College of Medicine, Taoyuan 33305, Taiwan; weger@cgmh.org.tw; 4Department of Thoracic Medicine, Chang Gung Memorial Hospital, Chang Gung University College of Medicine, Taoyuan 33305, Taiwan; ma7384@adm.cgmh.org.tw; 5Department of Thoracic Medicine, New Taipei Municipal TuCheng Hospital and Chang Gung University, Taoyuan 33302, Taiwan; 6Division of Thoracic Surgery, Department of Surgery, Taipei Veterans General Hospital, Taipei 112201, Taiwan; 7Graduate Institute of Clinical Medical Sciences, College of Medicine, Chang Gung University, Taoyuan 33302, Taiwan

**Keywords:** mechanical ventilation, acute respiratory distress syndrome, open lung biopsy, histology, diffuse alveolar damage, pulmonary fibrosis, idiopathic pulmonary fibrosis, outcomes

## Abstract

Background: Mechanical ventilation brings the risk of ventilator-induced lung injury, which can lead to pulmonary fibrosis and prolonged mechanical ventilation. Methods: A retrospective analysis of patients with acute respiratory distress syndrome (ARDS) who received open lung biopsy between March 2006 and December 2019. Results: A total of 68 ARDS patients receiving open lung biopsy with diffuse alveolar damage (DAD; the hallmark pathology of ARDS) were analyzed and stratified into non-fibrosis (*n* = 56) and fibrosis groups (*n* = 12). The duration of ventilator usage and time spent in the intensive care unit and hospital stay were all significantly higher in the fibrosis group. Hospital mortality was higher in the fibrosis than in the non-fibrosis group (67% vs. 57%, *p* = 0.748). A multivariable logistic regression model demonstrated that mechanical power at ARDS diagnosis and ARDS duration before biopsy were independently associated with histological fibrosis at open lung biopsy (odds ratio 1.493 (95% CI 1.014–2.200), *p* = 0.042; odds ratio 1.160 (95% CI 1.052–1.278), *p* = 0.003, respectively). Conclusions: Our findings indicate that prompt action aimed at staving off injurious mechanical stretching of lung parenchyma and subsequent progression to fibrosis may have a positive effect on clinical outcomes.

## 1. Introduction

Acute respiratory distress syndrome (ARDS) is a heterogeneous syndrome with complex pathophysiologic mechanisms characterized by severe hypoxemia and high mortality [1]. The pathogenesis of ARDS includes an exudative phase, a proliferative phase, and a fibrotic phase. Alveolar type II cell hyperplasia occurs after the initial exudative phase, resulting in the formation of resident fibroblasts and extracellular matrix. As disease progresses to the fibrotic phase, extensive basement membrane damage and inadequate or delayed reepithelialization can lead to the development of interstitial and intraalveolar fibrosis, which often requires prolonged mechanical ventilator support [2].

The typical histological hallmark of ARDS is diffuse alveolar damage (DAD) manifesting as hyaline membrane formation, lung edema, inflammation, and hemorrhage [3]. Roughly only half of the patients diagnosed with ARDS based on current criteria present evidence of DAD upon open lung biopsy (OLB) or autopsy [4,5,6,7]. Note, however, that not all ARDS patients progress to the fibrotic phase, as any imbalance in profibrotic or antifibrotic mediators could affect progression to the fibroproliferative phase [8]. In patients with ARDS, pulmonary fibrosis often leads to prolonged mechanical ventilation and poor clinical outcomes [9,10].

Idiopathic pulmonary fibrosis (IPF) is a chronic progressive fibrotic interstitial pneumonia with unknown etiology and poor prognosis. It is characterized by deterioration of the lung parenchyma structure and respiratory function [11]. Acute exacerbation of idiopathic pulmonary fibrosis (AE-IPF) has been defined as an acute clinically significant respiratory deterioration of unidentifiable cause, commonly leading to acute hypoxemia respiratory failure requiring mechanical ventilation [12]. The typical histological features of AE-IPF are DAD and/or organizing pneumonia superimposed on the usual interstitial pneumonia (UIP) pattern (i.e., acute lung injury occurring in an IPF/UIP lung), such that it shares many of the pathological features of ARDS [13,14].

A lung-protective ventilation strategy remains the cornerstone of treatment for ARDS patients and has been positively correlated with improved survival [2]; however, it brings with it the risk of ventilator-induced lung injury (VILI) and subsequent lung fibrosis [15]. Researchers have yet to establish optimal ventilator settings for patients with pulmonary fibrosis. Our objective in this study was to examine the association between serial changes in ventilator settings and histological fibrosis in ARDS patients with DAD based on OLB and compare clinical outcomes between fibrosis and non-fibrosis groups.

## 2. Materials and Methods

### 2.1. Study Design and Patients

This retrospective study was based on analysis of all ARDS patients who underwent OLB at Chang Gung Memorial Hospital (CGMH) in Taiwan between March 2006 and December 2019. CGMH is a tertiary care referral center with a 3700-bed general ward and 278-bed adult intensive care unit (ICU). The exclusion criteria included age of <20 years and histological findings not indicative of DAD.

At our institution, the decision to perform OLB was made by the treating intensivist in cases where the etiology of ARDS was unknown and the patient presented rapid pulmonary infiltration following a complete microbiologic examination including bronchoalveolar lavage. Surgical procedures were performed by a chest surgeon in an operating room or at the bedside in the ICU. Informed consent was obtained from the family prior to OLB. Under general anesthesia and mechanical ventilator support with adequate oxygenation, video-assisted thoracoscopic surgery or a 5-cm thoracotomy was used to secure the origins using an endoscopic stapler-cutter. The biopsy site was a new or progressive lesion identified via high-resolution computed tomography (HRCT) or chest X-ray. Every tissue specimen was examined by pathologists. The local Institutional Review Board for Human Research approved this study (CGMH IRB No. 202000760A3 and 202100595A3), and the need for informed consent was waived due to the retrospective nature of the study.

### 2.2. Definitions

ARDS was defined in accordance with the Berlin criteria [7]. Dynamic driving pressure (∆P) was calculated as peak inspiratory pressure (Ppeak) minus positive end-expiratory pressure (PEEP) [16]. Mechanical power (MP) was calculated using the following equation [17,18]: MP (Joules/minutes) (J/min) = 0.098 × tidal volume (V_T_) × respiratory rate (RR) × (Ppeak–1/2 × ∆P).

Ppeak is equivalent to plateau pressure in pressure-controlled ventilation, and Ppeak was used as a surrogate for plateau pressure to calculate MP if not specified [17]. Ventilator-free days was defined as the number of days between day 1 and day 28 or day 90 in which the patient breathed without assistance for at least 48 consecutive hours. Patients who did not survive to 28 days or 90 days were assigned zero ventilator-free days. Patients were stratified into the non-fibrosis group (DAD with exudative or proliferative phase) or the fibrosis group (DAD with fibrotic phase or chronic feature of honeycomb fibrosis).

### 2.3. Data Collections

Demographic data, comorbidities, and the etiologies of ARDS were recorded from hospital charts. The dates of hospital and ICU admission, mechanical ventilator initiation and liberation, date of ARDS diagnosis and OLB, ICU and hospital discharge, and time of death were collected. Arterial blood gas and mechanical ventilator settings parameters were recorded at approximately 10 a.m. daily after ARDS onset.

### 2.4. Histological Diagnosis

Based on analysis of hematoxylin- and eosin-stained lung tissue slices, a diagnosis of DAD (with or without fibrosis) was made independently by at least two pathologists who were blinded to the patients’ clinical information. Any discrepancies were discussed by the pathologists with the aim of reaching a final consensus as to histological diagnosis. The presence of DAD indicated hyaline membrane formation, intra-alveolar edema, alveolar type I cell necrosis, alveolar type II cell proliferation, the interstitial proliferation of myofibroblasts and fibroblasts, or organizing interstitial fibrosis [3]. AE-IPF represents DAD and/or organizing pneumonia superimposed on the UIP pattern [13,14]. We defined fibrosis as the manifestation of collagenous fibrosis or chronic appearance of microcystic honeycombing, or both [19].

### 2.5. Statistical Analysis

Continuous variables are presented as mean ± standard deviation or median (interquartile range), and categorical variables are reported as numbers (percentages). A student’s *t*-test or the Mann–Whitney *U* test were used to compare continuous variables among groups. Categorical variables were tested using the chi-square test for equal proportions or the Fisher’s exact test. Risk factors associated with histological fibrosis at OLB day were initially analyzed using univariate analysis, followed by a multivariable logistic regression model with stepwise selection. The results are presented using odds ratios and 95% confidence intervals (CIs). All statistical analysis was performed using SPSS 26.0 statistical software, and statistical significance was considered when the 2-sided *p* value was less than 0.05.

## 3. Results

### 3.1. Study Populations

This study identified 89 ARDS patients who underwent OLB within the study period and screened for inclusion and exclusion criteria. All ARDS patients were deeply sedated and paralyzed during the initial phase, including the day of OLB, and most cases received pressure-controlled ventilation until attempts at weaning from the mechanical ventilator. After excluding 21 patients who did not fulfill the histological DAD, 68 patients with histological DAD were analyzed (Figure 1). Based on the histological findings, 56 patients were assigned to the non-fibrosis group, and 12 patients were assigned to the fibrosis group. In the fibrosis group, 7 patients had DAD with the fibrotic phase and 5 patients presenting the AE-IPF.

### 3.2. Histological Findings

Compared to normal lung tissue (Figure 2A), DAD in the initial stage was characterized by the formation of hyaline membrane (Figure 2B) and in the later stage by interstitial fibrosis (Figure 2C). The term honeycomb fibrosis refers to the typical appearance of cysts in scarred lung tissue (i.e., UIP pattern) (Figure 2D). Organizing pneumonia was deemed indicative of acute exacerbation of UIP (Figure 2E).

### 3.3. Baseline Characteristics and Clinical Variables: Non-Fibrosis and Fibrosis Groups

As shown in Table 1, we observed no significant difference between non-fibrosis and fibrosis groups in terms of age, gender, body weight, body mass index, or comorbidities. In both groups, most of the ARDS cases were attributable to pulmonary causes (*n* = 58, 85%). The median interval between ARDS diagnosis and biopsy was significantly longer in the fibrosis group than the non-fibrosis group (18 (9–30) vs. 8 (5–12) days, *p* = 0.024). In terms of ventilator settings at the time of ARDS diagnosis, MP, Ppeak, and dynamic ∆P levels were significantly higher in the fibrosis group than in the non-fibrosis group (all *p* < 0.05). On the day of biopsy, Ppeak and dynamic ∆P values remained significantly higher in the fibrosis group than in the non-fibrosis group, and dynamic compliance was significantly lower (all *p* < 0.05).

The overall hospital mortality rate was 59%, and the mortality was relatively higher in the fibrosis group than in the non-fibrosis group (67% vs. 57%, *p* = 0.748). The duration of mechanical ventilation, length of ICU stay, and length of hospital stay were significantly higher in the fibrosis group than in the non-fibrosis group (all *p* < 0.05).

### 3.4. Baseline Characteristics and Clinical Variables: DAD with a Fibrotic Phase and AE-IPF Groups

As shown in Table 2, the patients in the AE-IPF group were older than those in the DAD with a fibrotic phase group. There were no significant differences between the two groups in terms of age, gender, body weight, body mass index, or comorbidities. In terms of ventilator settings at ARDS diagnosis, we observed no significant differences between the two groups. The median interval between ARDS diagnosis and biopsy was longer in the DAD with a fibrotic phase group than the AE-IPF group (28 (17–53) vs. 7 (4–21) days). On the day of biopsy, patients in the AE-IPF group received higher MP, higher V_T_, higher Ppeak, higher mean airway pressure, higher dynamic ∆P, and higher RR than patients in the DAD with a fibrotic phase group, although the difference did not reach the level of significance (Table 2 and Figure 3). Dynamic compliance was similar between the two groups on the day of biopsy.

### 3.5. Clinical Outcomes: DAD with a Fibrotic Phase and AE-IPF

As shown in Table 3 and Figure 4, 90-day hospital mortality was higher in the AE-IPF group than in the DAD with a fibrotic phase group (80% vs. 57%, *p* = 0.242). The duration of mechanical ventilation, length of ICU stay, and length of hospital stay were higher in the DAD with a fibrotic phase group than the AE-IPF group. The number of ventilator-free days at day 90, and ICU-free days at day 60 were also higher in the DAD with a fibrotic phase group than the AE-IPF group.

### 3.6. Factors Associated with Histological Fibrosis at OLB

After adjusting for significant confounding variables, a multivariable logistic regression model revealed that patients who had higher MP at ARDS diagnosis and had a longer ARDS duration before biopsy were significantly associated with histological fibrosis at OLB (odds ratio 1.493 (95% CI 1.014–2.200), *p* = 0.042 and odds ratio 1.160 (95% CI 1.052–1.278), *p* = 0.003, respectively) (Table 4).

## 4. Discussion

The primary insight in this study was the fact that ARDS patients with histological DAD and fibrosis based on OLB received significantly higher airway pressures, and had significantly longer durations of mechanical ventilator use and longer ICU and hospital stays. Hospital mortality was higher in the fibrosis group than in the non-fibrosis group. In the fibrosis group, patients with AE-IPF received higher ventilator load and had higher hospital mortality than those in the DAD with a fibrotic phase group.

DAD is the pathological hallmark of ARDS. In the current study, we enrolled only ARDS patients with histological evidence of DAD using data from OLB, unlike previous studies that included ARDS patients with and without DAD using data from OLB or autopsy [6,20,21]. The pathogenesis or the time course of lung damage in ARDS proceeds through an exudative phase (for roughly the first week after ARDS onset), proliferative phase (between the first and third weeks following ARDS onset), and fibrotic phase (beyond 3 or 4 weeks after ARDS onset) [2,22]. In the current study, the median duration from ARDS diagnosis to OLB was 28 days in the DAD with a fibrotic phase group. Some of the patients had progressed to DAD with a fibrotic phase within 3 weeks of ARDS onset. This indicates that the diagnosis of ARDS depends on clinical criteria, and that the onset of lung damage and histological fibrosis may begin before all the criteria for clinical diagnosis of ARDS are met [19]. Patients in the fibrosis group had significantly longer ARDS duration before biopsy (i.e., could have a longer duration of lung damage) than patients in the non-fibrosis group. In a multivariable logistic regression model, longer ARDS durations before biopsy were independently associated with histological fibrosis at OLB. This indicates that mechanical ventilation, histological fibrosis at OLB, and clinical outcomes of ARDS patients were related to the onset of lung damage.

The causes of pulmonary fibrosis during ARDS progression are multifactorial (e.g., inflammation and VILI) [9,15]. The course and onset of lung fibrosis can be traced to persistent injury and repair in response to mechanical strain and stress on epithelial cell resulting from volutrauma and atelectrauma, which subsequently triggers the fibroproliferative cascade [15,23]. The mechanical force could cause numerous intracellular mediators directly or indirectly released into the lung, which induces further lung damage and the subsequent development of lung fibrosis [15]. The extent of the ventilator load needed to cause pulmonary fibrosis was unknown. However, mechanical ventilation, histological fibrosis at OLB, and the clinical outcomes of ARDS patients could be directly related to the onset of lung fibrosis. MP refers to the energy delivered by a ventilator to the respiratory system per unit of time, as determined by volume, pressure, flow, and RR. Researchers have established that MP is of higher predictive value than individual ventilator parameters in assessing the risk of VILI [17,18]. Excessive MP has been shown to promote VILI and appears to be strongly correlated with histology (DAD score) and the expression of interleukin-6, a marker of inflammation [24]. Driving pressure is inversely proportional to lung compliance and aerated remaining functional lung size and has been linked to mortality in ARDS patients [16,25].

Few studies have examined the correlation between serial changes in ventilator settings and the development of histological fibrosis in ARDS patients. Serial changes in the ventilator settings may reflect the severity of nonresolving lung damage. In the current study, the MP and airway pressures (Ppeak and dynamic ∆P) received by the fibrosis group were significantly higher than those received by the non-fibrosis group at ARDS diagnosis. Thus, it is likely that the formation of lung fibrosis can be attributed at least in part to energy load (i.e., MP). In a multivariable logistic regression model, higher MP at ARDS diagnosis was independently associated with histological fibrosis at OLB. Compliance of the respiratory system was associated with the severity of lung injury, duration of ARDS, extent of lung fibrosis, and clinical outcomes [19]. Dynamic compliance in the fibrosis group was significantly lower and dropped more rapidly than in the non-fibrosis group on biopsy day. This may be due to the presence of fibrosis and a longer interval between ARDS diagnosis and OLB in the fibrosis group.

IPF is a form of chronic fibrosing interstitial pneumonia characterized by a progressive decline in lung function with radiological and/or histopathological indications of UIP [11,26]. Note, however, that UIP is not synonymous with IPF. The UIP pattern of fibrosis has also been linked to other conditions, such as connective tissue disease (mostly rheumatoid arthritis), drug toxicity, chronic hypersensitivity pneumonitis, asbestosis, and Hermansky–Pudlak syndrome [26,27]. AE-IPF and ARDS are quite similar in terms of DAD, lung inflammation, and respiratory mechanics. AE-IPF can lead to severe acute hypoxemic respiratory failure, requiring mechanical ventilator support. Patients with AE-IPF face a higher risk of mortality (may reach 95%) [14], which may be related to the fact that IPF patients tend to be older. In our study, 5 of the 12 patients in the fibrosis group presented histological findings indicative of UIP, and none of these patients presented with connective tissue disease, drug toxicity, or asbestosis. A pathologist and radiologist agreed that the cause of respiratory failure in this group was primarily AE-IPF. Patients in the AE-IPF group were older than those in the DAD with a fibrotic phase group (mean age 73.6 vs. 59.3 years), and the mortality was higher than DAD with a fibrotic phase group (80% vs. 57%).

Impaired lung mechanics due to structural, biochemical, and anatomical aberrations render fibrotic lungs susceptible to VILI [28]. At present, there is no solid evidence indicating the optimal and personalized ventilator settings for fibrotic lungs, including AE-IPF; some concepts can be derived from the evidence regarding ARDS because both share some common features. A “lung resting strategy” to avoid high PEEP during expiration and thereby prevent further lung injury has been posited as an alternative to the “open lung approach” (for ARDS cases presenting only DAD) for patients with pulmonary fibrosis and UIP, due to the fact that the presence of fibrotic tissue renders the lung structure highly fragile and prone to VILI (i.e., the “squishy ball lung” concept) [28]. In our study, patients in the DAD with a fibrotic phase and AE-IPF groups received similar ventilator settings except for V_T_ at ARDS diagnosis. However, patients in the AE-IPF group received a higher energy load (i.e., MP), higher V_T_, and higher airway pressures than in the DAD with a fibrotic phase group on the biopsy day. This indicates that intensivists may not recognize the disease status well and apply lung-protective ventilation at ARDS onset promptly; however, as disease progression, patients in the AE-IPF group received a higher ventilator load than those in the DAD with a fibrotic phase group on biopsy day due to underlying chronic fibrotic lungs, which contributed to a higher risk of VILI.

The strength of our study was that we investigated pulmonary fibrosis based on histological fibrosis from OLB. Previous studies examining the effect of pulmonary fibrosis on clinical outcomes in ARDS patients reported a link between HRCT scores indicative of fibroproliferative changes and clinical outcomes/mortality [10,29]. Nonetheless, thin-section CT scanning, including inspiratory, expiratory, and prone sequences, is the most important tool by which to evaluate pulmonary fibrosis. Serial images and quantitative estimates are also essential for differentiating fibrosis progression [27]. Overall, imaging alone cannot be relied upon to confirm destruction of the lung parenchyma, delineate active fibroproliferation, or assess the degree of lung fibrosis [8]. At present, there is no definitive biomarker for DAD, and HRCT findings are insufficient to differentiate DAD from DAD with organizing pneumonia, which was indicative of acute exacerbation of UIP [30]. The only way to confirm the presence of DAD is to obtain lung tissues via OLB or autopsy [20]. Unfortunately, cases that end in autopsy are very likely to be more severe than live cases, and autopsy series are unable to differentiate clinical outcomes or effects on mortality [19,20]. In the current study, we investigated the effect of pulmonary fibrosis on clinical outcomes by enrolling ARDS patients with histological DAD and fibrosis who had undergone OLB.

This retrospective study was hindered by a number of limitations. First, all patients were from a single tertiary care referral center over a long enrollment period. Furthermore, we focused on only cases of ARDS that had undergone OLB who fulfilled the histological DAD with fibrosis (i.e., fibrotic phase or AE-IPF) or not, which limited the number of recruited patients. Note that we opted not to exclude the 5 patients with chronic fibrosis (i.e., UIP pattern) from the fibrosis group (*n* = 12), similar to that of a previous ARDS study based on autopsies in which half of the fibrosis group (15 of 30 patients) also presented with chronic microcystic honeycombing [19]. Besides, we further divided the fibrosis group into the DAD with a fibrotic phase or the AE-IPF group and compared clinical outcomes. Second, the causes of pulmonary fibrosis are complex and multifactorial, and the exact causal relationship between mechanical ventilation and pulmonary fibrosis was difficult to determine due to the retrospective nature of the study. It is important to emphasize that mechanical ventilation, histological fibrosis at OLB, and clinical outcomes of ARDS patients could be directly related to the onset of lung damage and fibrosis. Third, compliance with lung-protective ventilation with lower tidal volumes tends to decrease in real-world clinical practice [1]. Our study was conducted over a long study period from 2006 to 2019 with retrospective analysis, and there was no standard protocol for the ventilator settings among the enrolled ICUs. Therefore, the ARDS patients included in this study received a relatively high V_T_ than 6 mL/kg PBW of the current guidelines [2], which may make external validation to other ARDS cohorts difficult to perform and may have influenced the clinical outcomes. Finally, corticosteroids have anti-inflammatory and antifibrosis effects; however, we opted not to address the use of steroid therapy, due to a lack of evidence pertaining to the benefits of steroid treatment for persistent ARDS and fibrotic lungs.

## 5. Conclusions

Our findings revealed that ARDS patients with histological DAD and fibrosis received significantly higher airway pressures, underwent mechanical ventilation for a longer duration, and remained in the ICU and hospital for a longer period. Hospital mortality was higher in the fibrosis group than in the non-fibrosis group. In the fibrosis group, patients with AE-IPF received a higher ventilator load and faced higher mortality than those in DAD with a fibrotic phase.

Implementing optimal ventilator settings as early as possible may be necessary to reduce the risk of VILI and pulmonary fibrosis. Further large-scale studies are required to identify the mechanisms by which mechanical ventilation induces pulmonary fibrosis, and define safety standards aimed at minimizing the risk of VILI and fibrosis.

## Figures and Tables

**Figure 1 jpm-12-00474-f001:**
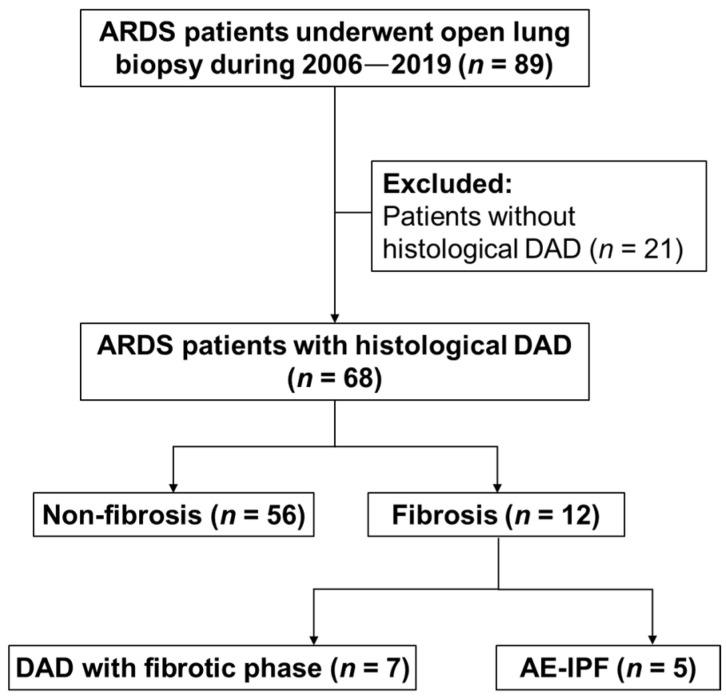
Flow chart of ARDS patients receiving open lung biopsy. ARDS: acute respiratory distress syndrome; DAD: diffuse alveolar damage; AE-IPF: acute exacerbation of idiopathic pulmonary fibrosis.

**Figure 2 jpm-12-00474-f002:**
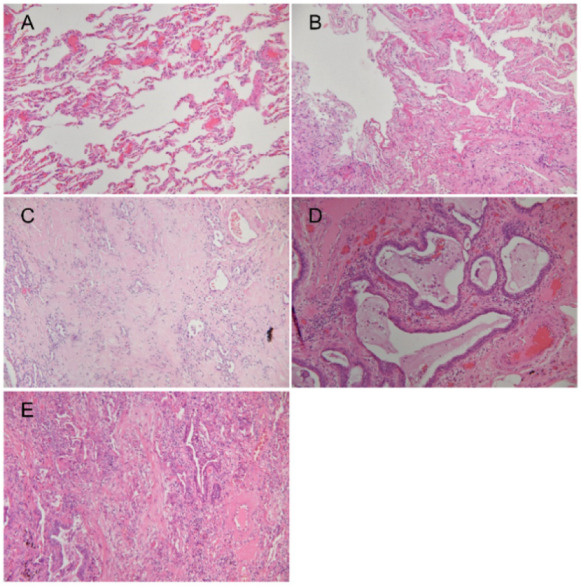
Human lung tissues samples from enrolled patients stained using hematoxylin and eosin. (**A**) Normal lung tissue defined as thin alveolar capillary membrane with clear alveolar space; (**B**) diffuse alveolar damage in which alveolar surfaces are lined with hyaline membranes; (**C**) lung tissue showing distinct indications of interstitial fibrosis; (**D**) UIP showing honeycomb fibrosis; (**E**) organizing pneumonia indicating acute exacerbation of UIP. 4X magnification. UIP: usual interstitial pneumonia.

**Figure 3 jpm-12-00474-f003:**
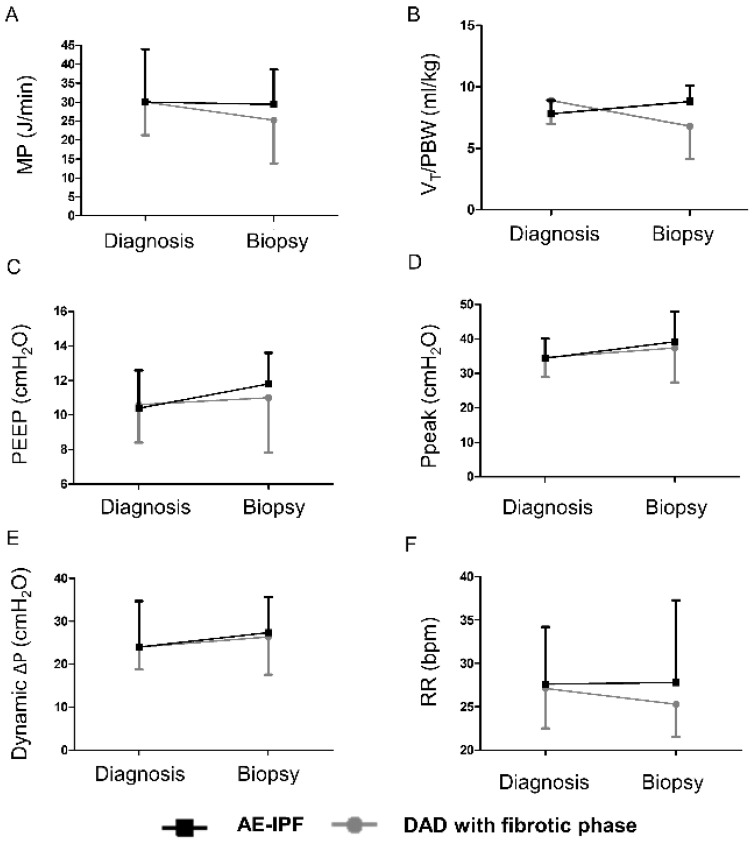
Serial changes in the ventilatory variables of (**A**) MP (**B**) V_T_/PBW (**C**) PEEP (**D**) Ppeak (**E**) Dynamic ∆P (**F**) RR between ARDS diagnosis and open lung biopsy of patients in the DAD with a fibrotic phase and AE-IPF groups. ARDS: acute respiratory distress syndrome; DAD: diffuse alveolar damage; AE-IPF: acute exacerbation of idiopathic pulmonary fibrosis; MP: mechanical power; V_T_: tidal volume; PBW: predicted body weight; PEEP: positive end-expiratory pressure; Ppeak: peak inspiratory pressure; ∆P: driving pressure; RR: respiratory rate; bpm: beats per minute.

**Figure 4 jpm-12-00474-f004:**
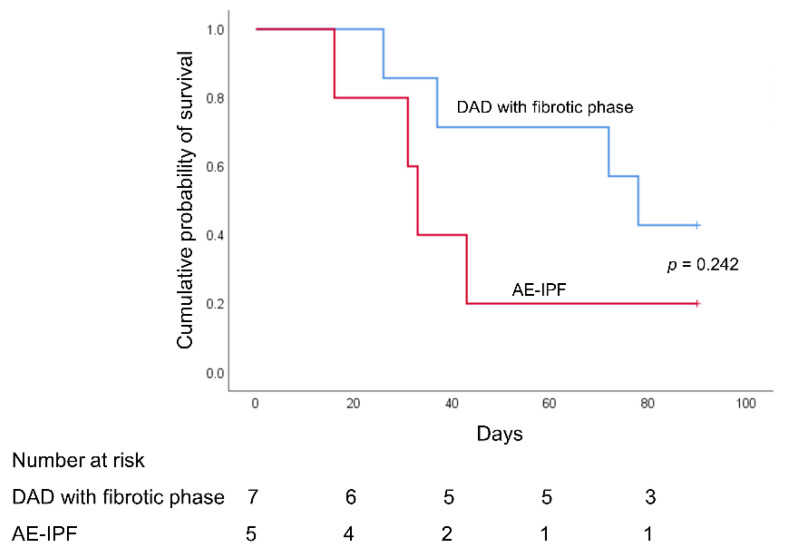
Kaplan–Meier 90-day survival curves of patients with acute respiratory distress syndrome undergoing open lung biopsy, as stratified by DAD with a fibrotic phase and AE-IPF. DAD: diffuse alveolar damage; AE-IPF: acute exacerbation of idiopathic pulmonary fibrosis.

**Table 1 jpm-12-00474-t001:** Background characteristics and clinical variables: non-fibrosis and fibrosis on histological findings.

Characteristics	All (*n* = 68)	Non-Fibrosis (*n* = 56)	Fibrosis (*n* = 12)	*p*
Age (years)	60.4 ± 16	59.4 ± 14.8	65.3 ± 20.8	0.255
Male (gender)	39 (57%)	31 (55%)	8 (67%)	0.472
Body weight (kg)	60.7 ± 11.7	61.0 ± 12.3	58.4 ± 6.1	0.566
Body mass index (kg/m^2^)	23.7 ± 3.8	23.8 ± 3.9	22.8 ± 3.3	0.488
Comorbidities				
Diabetes mellitus	12 (18%)	9 (16%)	3 (25%)	0.432
Hypertension	20 (29%)	18 (32%)	2 (17%)	0.486
Chronic lung diseases	7 (10%)	6 (11%)	1 (8%)	1.0
Immunocompromised	18 (27%)	16 (29%)	2 (17%)	0.494
ARDS etiologies				
Pulmonary causes	58 (85%)	47 (84%)	11 (92%)	0.678
Extrapulmonary causes	10 (15%)	9 (16%)	1 (8%)	0.678
PaO_2_/FiO_2_ at day of ARDS diagnosis (mmHg)	135 (61–204)	136 (61–213)	118 (58–187)	0.613
Ventilator settings at day of ARDS diagnosis				
Mechanical power (J/min)	24.6 ± 9	23.5 ± 8.2	29.9 ± 10.8	0.023
Tidal volume (mL/kg PBW)	8.6 ± 1.9	8.7 ± 2	8.4 ± 1.6	0.764
PEEP (cm H_2_O)	10.9 ± 2.4	11 ± 2.5	11 ± 2.1	0.566
Peak inspiratory pressure (cm H_2_O)	31.7 ± 5.6	31.1 ± 4.9	36.1 ± 5.3	0.004
Mean airway pressure (cm H_2_O)	17.6 ± 3.5	17.5 ± 3.4	18.5 ± 4	0.345
Dynamic driving pressure (cm H_2_O)	20.9 ± 5.2	20.2 ± 4.4	24 ± 7.5	0.021
Total respiratory rate (breaths/min)	24.9 ±5	24.3 ± 4.9	27.3 ± 5.2	0.088
Dynamic compliance (mL/cm H_2_O)	24.8 ± 10.5	24.8 ± 8.9	24.8 ± 16.5	0.989
Day from ARDS diagnosis to biopsy	8 (5–14)	8 (5–12)	18 (9–30)	0.024
PaO_2_/FiO_2_ at biopsy day (mmHg)	139 (104–194)	137 (103–199)	141 (98–191)	0.867
Ventilator settings at biopsy day				
Mechanical power (J/min)	23.8 ± 7.7	23.1 ± 6.9	27 ± 10.3	0.12
Tidal volume (mL/kg PBW)	7.7 ± 1.8	7.7 ± 1.7	7.6 ± 2.3	0.961
PEEP (cm H_2_O)	12.3 ± 2.6	12.5 ± 2.5	11.3 ± 2.6	0.191
Peak inspiratory pressure (cm H_2_O)	34.4 ± 6.9	33.6 ± 6.2	38.2 ± 9.1	0.037
Mean airway pressure (cm H_2_O)	19.3 ± 3.8	19.4 ± 3.6	18.8 ± 5.1	0.567
Dynamic driving pressure (cm H_2_O)	22.1 ± 6.8	21.1 ± 6.0	26.8 ± 8.2	0.007
Total respiratory rate (breaths/min)	25.2 ± 5	24.9 ± 4.7	26.3 ± 6.5	0.373
Dynamic compliance (mL/cm H_2_O)	20.5 ± 7.6	21.3 ± 7.6	17.1 ± 6.7	0.017
Hospital mortality, n (%)	40 (59%)	32 (57%)	8 (67%)	0.748
Duration of mechanical ventilator (days)	22 (15–34)	21 (14–32)	35 (24–74)	0.028
Length of ICU stay (days)	27 (17–37)	25 (16–34)	51 (32–80)	0.001
Length of hospital stay (days)	34 (22–56)	31 (20–46)	55 (32–81)	0.004
Ventilator-free days at day 28	0 (0–5)	0 (0–11)	0 (0–0)	0.036
ICU-free days at day 28	0 (0–7)	0 (0–8)	0 (0–0)	0.036
ICU-free days at day 60	0 (0–35)	0 (0–40)	0 (0–0)	0.008
Hospital-free days at day 90	51 (29–66)	54 (16–34)	25 (2–54)	0.01

Data is presented as mean ± standard deviation, count or median (interquartile range). ARDS: acute respiratory distress syndrome; PaO_2_: partial pressure of oxygen in arterial blood; FiO_2_: fraction of inspired oxygen; PBW: predicted body weight; PEEP: positive end-expiratory pressure; ICU: intensive care unit.

**Table 2 jpm-12-00474-t002:** Background characteristics and clinical variables: DAD with a fibrotic phase and AE-IPF on histological findings.

Characteristics	DAD with a Fibrotic Phase (*n* = 7)	AE-IPF (*n* = 5)	*p*
Age (years)	59.3 ± 25.5	73.6 ± 8.36	0.204
Male (gender)	5 (71%)	3 (60%)	0.769
Body weight (kg)	60.5 ± 8.1	56.4 ± 3.2	0.379
Body mass index (kg/m^2^)	23.4 ± 4.4	22.1 ± 1.7	0.657
Comorbidities			
Diabetes mellitus	1 (14%)	2 (40%)	0.31
Hypertension	1 (14%)	1 (20%)	0.793
Chronic lung diseases	1 (14%)	0 (0%)	0.377
Immunocompromised	2 (29%)	0 (0%)	0.19
ARDS etiologies			
Pulmonary causes	6 (86%)	5 (100%)	1.0
Extrapulmonary causes	1 (14%)	0 (0%)	1.0
PaO_2_/FiO_2_ at day of ARDS diagnosis (mmHg)	68 (39–138)	136 (118–176)	0.794
Ventilator settings at day of ARDS diagnosis			
Mechanical power (J/min)	30 ± 8.7	30 ± 14	0.987
Tidal volume (mL/kg PBW)	8.9 ± 1.9	7.8 ± 1.1	0.434
PEEP (cm H_2_O)	10.6 ± 2.2	10.4 ± 2.2	0.897
Peak inspiratory pressure (cm H_2_O)	34.6 ± 5.6	34.4 ± 5.7	0.971
Mean airway pressure (cm H_2_O)	18.6 ± 4.1	18.4 ± 4.4	0.946
Dynamic driving pressure (cm H_2_O)	24 ± 5.2	24 ± 10.7	1.0
Total respiratory rate (breaths/min)	27.1 ± 4.6	27.6 ± 6.6	0.889
Dynamic compliance (mL/cm H_2_O)	21.2 (15–24.6)	15.9 (14.1–49)	0.755
Day from ARDS diagnosis to biopsy	28 (17–53)	7 (4–21)	0.242
PaO_2_/FiO_2_ at biopsy day (mmHg)	168 (95–192)	106 (89–141)	0.546
Ventilator settings at biopsy day			
Mechanical power (J/min)	25.2 ± 11.4	29.4 ± 9.2	0.851
Tidal volume (mL/kg PBW)	6.8 ± 2.7	8.8 ± 1.3	0.301
PEEP (cm H_2_O)	11 ± 3.2	11.8 ± 1.8	0.394
Peak inspiratory pressure (cm H_2_O)	37.4 ± 10	39.2 ± 8.8	0.757
Mean airway pressure (cm H_2_O)	17.7 ± 5.3	20.4 ± 4.9	0.628
Dynamic driving pressure (cm H_2_O)	26.4 ± 8.8	27.4 ± 8.3	0.663
Total respiratory rate (breaths/min)	25.3 ± 3.8	27.8 ± 9.5	0.537
Dynamic compliance (mL/ cm H_2_O)	15.8 (11.7–18.6)	15.2 (10.1–27.6)	0.876

Data is presented as mean ± standard deviation, count, or median (interquartile range). DAD: diffuse alveolar damage; AE-IPF: acute exacerbation of idiopathic pulmonary fibrosis; ARDS: acute respiratory distress syndrome; PaO_2_: partial pressure of oxygen in arterial blood; FiO_2_: fraction of inspired oxygen; PBW: predicted body weight; PEEP: positive end-expiratory pressure.

**Table 3 jpm-12-00474-t003:** Clinical outcomes as a function of DAD with a fibrotic phase and AE-IPF on histological findings.

Outcomes	DAD with Fibrotic Phase (*n* = 7)	AE-IPF (*n* = 5)	*p*
90-day hospital mortality, *n* (%)	4 (57%)	4 (80%)	0.242
Other outcomes			
Duration of mechanical ventilator (days)	46 (22–75)	42 (24–66)	0.84
Length of ICU stay (days)	72 (37–81)	48 (24–78)	0.319
Length of hospital stay (days)	82 (53–93)	57 (27–98)	0.364
Ventilator-free days at day 90	14 (0–23)	0 (0–0)	0.224
ICU-free days at day 60	1 (0–0)	0 (0–0)	0.424

Data is presented as a count or median (interquartile range). DAD: diffuse alveolar damage; AE-IPF: acute exacerbation of idiopathic pulmonary fibrosis; ICU: intensive care unit.

**Table 4 jpm-12-00474-t004:** Factors associated with histological fibrosis at open lung biopsy using a multivariable logistic regression model.

Characteristics	Odds Ratio (95% CI)	*p*
MP at day of ARDS diagnosis (J/min)	1.493 (1.014–2.200)	0.042
ARDS duration before biopsy (days)	1.160 (1.052–1.278)	0.003

CI: confidence interval; MP: mechanical power; ARDS: acute respiratory distress syndrome. The multivariable analysis model included continuous variables (age, body weight, body mass index, ventilatory variables at day of ARDS diagnosis, and day from ARDS diagnosis to biopsy) and categorical variables (gender, comorbidities, and ARDS etiologies). For the continuous variables, the odds ratio means that the risk of histological fibrosis at open lung biopsy increases or decreases per unit increase in these variables.

## Data Availability

All data is available from the corresponding authors on reasonable request.

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
