# Peer review of "Relationship between Mechanical Ventilation and Histological Fibrosis in Patients with Acute Respiratory Distress Syndrome Undergoing Open Lung Biopsy"

_jpm, 2022, doi:10.3390/jpm12030474_

Round 1

Reviewer 1 Report

thanks for the opportunity to review this manuscript titled "Relationship between Mechanical Ventilation and Histological Fibrosis in Patients with Acute Respiratory Distress Syndrome Undergoing Open Lung Biopsy".

The study describes the mechanical stretching of lung parenchyma and subsequent progression to fibrosis in a retrospective population of patients admitted in ICU invasively mechanically ventilated undergoing OLB.

the study is interesting and clearly written however the number of patients is small due to the single center considered. Issues I would like to clarify are

  1. in table 1 the Tidal volume ventilation was in both group above the protective tidal volume ventilation of 6 ml/kg.  please explain why a protective ventilation mode was not use in these population of ARDS patients despite current guidelines
  2. This issue should be emphasized in the discussion as it could be directly related to the onset of lung damages and fibrosis.

Author Response

thanks for the opportunity to review this manuscript titled "Relationship between Mechanical Ventilation and Histological Fibrosis in Patients with Acute Respiratory Distress Syndrome Undergoing Open Lung Biopsy".

The study describes the mechanical stretching of lung parenchyma and subsequent progression to fibrosis in a retrospective population of patients admitted in ICU invasively mechanically ventilated undergoing OLB.

the study is interesting and clearly written however the number of patients is small due to the single center considered. Issues I would like to clarify are

Point 1: in table 1 the Tidal volume ventilation was in both group above the protective tidal volume ventilation of 6 ml/kg.  please explain why a protective ventilation mode was not use in these population of ARDS patients despite current guidelines

Response 1:

We thank the reviewer’s comment to point out this problem. A lung-protective ventilation strategy with lower tidal volumes and lower airway pressures remained the mainstay of management for acute respiratory distress syndrome (ARDS) [1–3]. This is a retrospective study and was analyzed a period of up to 13 years (2006–2019). There was no standard protocol for ventilator settings among the enrolled ICUs, and the recruited ARDS patients received higher tidal volume than recommended 6 ml/kg predicted body weight (PBW) of current guidelines.

Most ARDS patients did not receive low tidal volume ventilation in real-world clinical practice, and the mean tidal volume was 7.6 ml/kg PBW in the LUNG SAFE study [4]. In the present study, the tidal volume was 8.6 ml/kg PBW at day of ARDS diagnosis and 7.7 ml/kg PBW at biopsy day. Both the values exceeded the recommended 6 ml/kg PBW but were far below levels deemed injurious (12 ml/kg PBW) [3]. These may make external validation of our study to other ARDS cohorts problematic to perform.

We addressed the above limitations in the eighth paragraph of Discussion section in the revised manuscript as follows (marked with red text):

Eighth paragraph of Discussion:

…...Third, compliance with lung-protective ventilation with lower tidal volumes tends to drop in real-world clinical practice. Our study was conducted over the long study period from 2006 to 2019 with retrospective analysis, and there was no standard protocol for ventilator settings among the enrolled ICUs. Therefore, the ARDS patients included in this study received relatively high VT than 6 ml/kg PBW of current guidelines which may make external validation to other ARDS cohorts difficult to perform and may have influenced clinical outcomes.

Reference

  1. Thompson, B.T.; Chambers, R.C.; Liu, K.D. Acute Respiratory Distress Syndrome. N Engl J Med 2017, 377, 562–
  2. Fan, E.; Del, Sorbo. L.; Goligher, E.C.; Hodgson, C.L.; Munshi, L.; Walkey, A.J.; Adhikari, N.K.J; Amato, M.B.P; Branson, R.; Brower, R.G.; et al.; American Thoracic Society, European Society of Intensive Care Medicine, and Society of Critical Care Medicine. An Official American Thoracic Society/European Society of Intensive Care Medicine/Society of Critical Care Medicine Clinical Practice Guideline: Mechanical Ventilation in Adult Patients with Acute Respiratory Distress Syndrome. Am J Respir Crit Care Med 2017, 195, 1253–1263.
  3. Acute Respiratory Distress Syndrome Network, Brower, R.G.; Matthay, M.A.; Morris, A.; Schoenfeld, D.; Thompson, B.T.; Wheeler, A. Ventilation with lower tidal volumes as compared with traditional tidal volumes for acute lung injury and the acute respiratory distress syndrome. N Engl J Med 2000, 342, 1301–8.
  4. Bellani, G.; Laffey, J.G.; Pham, T.; Fan, E.; Brochard, L.; Esteban, A.; Gattinoni, L.; van Haren, F.; Larsson, A.; McAuley, D.F.; et al. LUNG SAFE Investigators; ESICM Trials Group. Epidemiology, Patterns of Care, and Mortality for Patients with Acute Respiratory Distress Syndrome in Intensive Care Units in 50 Countries. JAMA 2016, 315, 788–800.

Point 2: This issue should be emphasized in the discussion as it could be directly related to the onset of lung damages and fibrosis.

Response 2:

We thank the reviewer’s suggestion and this is an excellent point of view. We agreed that this issue could be directly related to the onset of lung damages and fibrosis.

The pathogenesis or the time course of lung damages of ARDS proceeds through an exudative phase (for roughly the first week after ARDS onset), proliferative phase (between the first- and third-weeks following ARDS onset), and fibrotic phase (beyond 3 or 4 weeks after ARDS onset). The onset of lung damages (i.e., probable the onset of ARDS) and fibrosis may not be detected easily and could begin before clinical criteria of ARDS met.

In the current study, patients in the fibrosis group had significantly longer ARDS duration before biopsy (i.e., could have longer duration of lung damages) than did patients in the non-fibrosis group (18 [9–30] vs. 8 [5–12] days, p = 0.024). A multivariable logistic regression model revealed that longer ARDS duration before biopsy remained independently associated with histological fibrosis at open lung biopsy (odds ratio 1.160 [95% CI 1.052–1.278], p = 0.003).

 Besides, patients in the fibrosis group received significantly higher airway pressures than the non-fibrosis group (p < 0.05). The mortality was relatively higher in the fibrosis group than in the non-fibrosis group (67% vs. 57%, p = 0.748). The duration of mechanical ventilation, length of ICU stay, and length of hospital stay were significantly higher in the fibrosis group than in the non-fibrosis group (all p < 0.05). It indicated that mechanical ventilation, histological fibrosis at open lung biopsy, and clinical outcomes of ARDS patients could be directly related to the onset of lung damages.

The causes of pulmonary fibrosis during ARDS could be related to inflammation, ventilator-induced lung injury (VILI) or other risk factors [1, 2]. The onset of lung fibrosis in ARDS can be traced to persistent injury and repair in response to mechanical strain and stress on epithelial cell which triggers the fibroproliferative cascade subsequently [3]. The mechanical force could cause numerous intracellular mediators directly or indirectly released into the lung which induced further lung damages and subsequent development of lung fibrosis [2, 4]. Pulmonary fibrosis could lead to prolonged mechanical ventilation and poor clinical outcomes [1]. The extent of ventilator load needed to cause pulmonary fibrosis was unknown. However, mechanical ventilation, histological fibrosis at open lung biopsy, and clinical outcomes of ARDS patients could be directly related to the onset of fibrosis.

We addressed the above description in the second, third, and eighth paragraph of Discussion section in the revised manuscript as follows (marked with red text):

Second paragraph of Discussion:

…...This indicates that the diagnosis of ARDS depends on clinical criteria, and that the onset of lung damages and histological fibrosis may begin before all the criteria for clinical diagnosis of ARDS are me. Patients in the fibrosis group had significantly longer ARDS duration before biopsy (i.e., could have longer duration of lung damages) than did patients in the non-fibrosis group. In a multivariable logistic regression model, longer ARDS duration before biopsy were independently associated with histological fibrosis at OLB. It indicated that mechanical ventilation, histological fibrosis at OLB and clinical outcomes of ARDS patients were related to the onset of lung damages.

Third paragraph of Discussion:

…...The causes of pulmonary fibrosis during ARDS progression are multifactorial (e.g., inflammation and VILI). The course and onset of lung fibrosis can be traced to persistent injury and repair in response to mechanical strain and stress on epithelial cell resulting from volutrauma and atelectrauma which triggers the fibroproliferative cascade subsequently. The mechanical force could cause numerous intracellular mediators directly or indirectly released into the lung which induced further lung damages and subsequent development of lung fibrosis. The extent of ventilator load needed to cause pulmonary fibrosis was unknown. However, mechanical ventilation, histological fibrosis at OLB and clinical outcomes of ARDS patients could be directly related to the onset of lung fibrosis.

Eighth paragraph of Discussion:

……It is important to emphasize that mechanical ventilation, histological fibrosis at OLB, and clinical outcomes of ARDS patients could be directly related to the onset of lung damages and fibrosis.

Reference

  1. Cabrera-Benitez, N.E.; Laffey, J.G.; Parotto, M.; Spieth, P.M.; Villar, J.; Zhang, H.; Slutsky, A.S. Mechanical ventilation-associated lung fibrosis in acute respiratory distress syndrome: a significant contributor to poor outcome. Anesthesiology 2014, 121, 189–
  2. Slutsky, A.S.; Ranieri, V.M. Ventilator-induced lung injury. N Engl J Med 2013, 369, 2126–
  3. Michalski, J.E.; Kurche, J.S.; Schwartz, D.A. From ARDS to pulmonary fibrosis: the next phase of the COVID-19 pandemic? Transl Res. 2022, 241, 13–24.
  4. Albert, R.K.; Smith, B.; Perlman, C.E.; Schwartz, D.A. Is Progression of Pulmonary Fibrosis due to Ventilation-induced Lung Injury? Am J Respir Crit Care Med 2019, 200, 140–

We thank the reviewer for valuable comments. Addressing them fully has significantly strengthened the manuscript.

Reviewer 2 Report

Congratulations to the authors for this excellent study. The results of the study  have  a significant impact  on in clinical practice  As is   proved the significance on injurious ventilation on patients outcome.   Some limitations are well listed and discused ny the authors in the duscussion section. 
I have no  other comments 

Author Response

Congratulations to the authors for this excellent study. The results of the study  have  a significant impact  on in clinical practice  As is   proved the significance on injurious ventilation on patients outcome.   Some limitations are well listed and discused ny the authors in the duscussion section. 
I have no  other comments 

Response:

We thank the reviewer’s comment and appreciation. Our results reported the significance of injurious ventilation on clinical outcomes of ARDS patients with histological fibrosis undergoing open lung biopsy, and it may have a significant impact on clinical practice.

We thank the reviewer for valuable comments. Addressing them fully has significantly strengthened the manuscript.
